# Role of immunosuppression in an antibiotic stewardship intervention and its association with clinical outcomes and antibiotic use: protocol for an observational study (RISC-sepsis)

Jonathan Scott,[1] Loredana Trevi,[1] Hannah McNeil,[2] Tom Ewen,[1] Phil Mawson,[1] David McDonald,[3] Andrew Filby,[3] Ranjit Lall,[2] Katie Booth,[2] Gert Boschman,[4] Vesna Melkebeek,[4] Gavin Perkins,[2,5] Ronan McMullan,[6,7] Daniel F McAuley [7,8] Iain J McCullagh,[1,9] Timothy Walsh,[10,11] Anthony Rostron,[1,12] Manu Shankar-Hari [10,11] Paul Dark,[13,14] A John Simpson,[1,15] Andrew Conway Morris,[16,17,18] Thomas P Hellyer [1,19]

JS and LT contributed equally.

For numbered affiliations see end of article.

**Correspondence to**
Dr Thomas P Hellyer;
thomas.hellyer@newcastle.ac.uk

## ABSTRACT

**Introduction** Sepsis is characterised by a dysregulated immune response to infection, with exaggerated pro-inflammatory and anti-inflammatory responses. A predominant immunosuppressive profile affecting both innate and adaptive immune responses is associated with increased hospital-acquired infection and reduced infection-free survival. While hospital-acquired infection leads to additional antibiotic use, the role of the immunosuppressive phenotype in guiding complex decisions, such as those affecting antibiotic stewardship, is uncertain. This study is a mechanistic substudy embedded within a multicentre clinical and cost-effectiveness trial of biomarker-guided antibiotic stewardship. This mechanistic study aims to determine the effect of sepsis-associated immunosuppression on the trial outcome measures.

**Methods and analysis** RISC-sepsis is a prospective, multicentre, exploratory, observational study embedded within the ADAPT-sepsis trial. A subgroup of 180 participants with antibiotics commenced for suspected sepsis, enrolled in the ADAPT-sepsis trial, will be recruited. Blood samples will be collected on alternate days until day 7. At each time point, blood will be collected for flow cytometric analysis into cell preservation tubes. Immunophenotyping will be performed at a central testing hub by flow cytometry. The primary outcome measures are monocyte human leucocyte antigen-DR; neutrophil CD88; programmed cell death-1 on monocytes, neutrophils and T lymphocytes and the percentage of regulatory T cells. Secondary outcome measures will link to trial outcomes from the ADAPT-sepsis trial including antibiotic days; occurrence of hospital-acquired infection and length of ICU-stay and hospital-stay.

**Ethics and dissemination** Ethical approval has been granted (IRAS 209815) and RISC-sepsis is registered with the ISRCTN (86837685). Study results will be disseminated by peer-reviewed publications, presentations at scientific meetings and via patient and public participation groups and social media.

## STRENGTHS AND LIMITATIONS OF THIS STUDY

⇒ Multicentre study recruiting a broad cohort of patients, representative of critically ill patients with sepsis in the UK.
⇒ Centralised flow cytometry immunophenotyping using preservation methods allowing standardisation in the context of multicentre recruitment.
⇒ Embedded within a clinical trial, immunophenotypes will be linked to robust clinical outcomes.
⇒ As an observational study, RISC-sepsis will provide insights into the impact of sepsis-associated immunosuppression on a biomarker-guided antibiotic duration intervention but since it is not interventional, it will not offer a robust precision-medicine approach to antibiotic stewardship.

## INTRODUCTION

Sepsis is a syndrome of life-threatening organ dysfunction secondary to a dysregulated immune response to an infection.[1] Mortality from sepsis is high, with 31% of patients dying in hospital and a further 15% of survivors dying in the subsequent year.[2,3] As a leading cause of death, there are an estimated 31.5 million cases of sepsis worldwide each year.[4] Organ support in a critical care setting is often required and patients with sepsis account for a third of general adult critical care admissions in the UK.[2]

The dysregulated immune response that characterises sepsis is a paradigm of both a pro-inflammatory and an anti-inflammatory response.[5,6] While these immune states can occur simultaneously, a predominant immunosuppressive profile has been associated

with hospital-acquired infections and mortality.[7 8] While there is no single test for sepsis-associated immunosuppression, well-established leucocyte cell surface markers identify cellular dysfunction and are associated with adverse clinical outcomes. These dysfunctions are demonstrated across both the innate and adaptive immune responses with the most well-established markers being monocyte human leucocyte antigen-DR (mHLA-DR), neutrophil C5a receptor (CD88), percentage of regulator T cells and programmed cell death-1 (PD-1).[9–12] The persistence of these cellular abnormalities and the accumulation of dysfunctions increase the risk of adverse clinical outcomes.[10 13–15]

While an immunosuppressive profile has been linked to poor clinical outcomes, the influence that it has in guiding complex clinical decision-making processes, such as those involved in antibiotic stewardship remain uncertain. Antibiotic treatment in sepsis may be life-saving, but the overuse of antibiotics leads to the emergence of antimicrobial resistance (AMR) and adverse clinical outcomes.[16–19] The optimal duration of antibiotic treatment in sepsis is unknown.[20] Biomarker-guided antibiotic durations are not recommended for routine use in the UK[21] and the current evidence for effectiveness is regarded as low quality internationally.[22] The ADAPT-Sepsis trial (ISRCTN47473244) is a UK National Institute for Health and Care Research (NIHR) Health Technology Assessment-commissioned clinical trial that will evaluate the clinical and cost-effectiveness of procalcitonin (PCT) and C reactive protein (CRP) protocols for biomarker-guided antibiotic duration decisions in a National Health Service setting. This three-arm trial of daily PCT monitoring, daily CRP monitoring or standard of care aims to determine whether the use of these biomarkers reduces total antibiotic use at 28 days (primary superiority outcome), while maintaining treatment safety of 28-day mortality (primary non-inferiority outcome). Investigations into the dynamics of CRP and PCT with mHLA-DR have shown that high levels of CRP or PCT are associated with low expression of mHLA-DR,[23–25] suggesting CRP and PCT offer a window into the more complex dysregulation of the immune response.

This protocol outlines a mechanistic study embedded within the ADAPT-Sepsis trial. RISC-sepsis will immune phenotype patients using leucocyte surface biomarkers and dichotomise patients into two groups, 'sepsis-associated immunosuppression' and 'non-immunosuppressed' (definition below). We will then explore the effectiveness of the biomarker-guided antibiotic duration intervention and the impact on trial outcomes for patients with sepsis-associated immunosuppression.

## Rationale

The ADAPT-Sepsis trial will include a heterogeneous population of adult patients with sepsis, including those with sepsis-induced immunosuppression. As a phenotype associated with hospital-acquired infections and reduced infection-free survival, there may be heterogeneity in the effect of the biomarker-guided antibiotic duration intervention for patients with this phenotype. If biomarker-guided antibiotics are ineffective in patients with sepsis-associated immunosuppressed, then a greater degree of patient stratification may be required to ensure optimal antibiotic duration. Furthermore, if CRP and PCT levels fall over the same period that cellular defects resolve, this may give clinicians greater confidence in following biomarker-guided antibiotic durations.

## Research question

Does sepsis-associated immunosuppression render a PCT/CRP biomarker-guided antibiotic duration intervention ineffective?

## Hypothesis

Sepsis-associated immunosuppression will be associated with longer duration of antibiotics, due to persistently raised PCT and CRP; more hospital-acquired infection and more antibiotic treatment days.

## Primary objective

To determine differences in ADAPT-sepsis trial outcomes between patients defined as 'sepsis-associated immunosuppressed' versus 'non-immunosuppressed'.

## Secondary objective

To monitor changes in expression of leucocyte surface markers (primary outcome measures) over the time course of the ADAPT-Sepsis biomarker-guided antibiotic duration intervention period.

## METHODS AND ANALYSIS

RISC-sepsis is a prospective, multicentre, exploratory, observational study embedded within the ADAPT-Sepsis trial.

## Population

We will include sequential patients who are enrolled into the ADAPT-sepsis trial within sites in the RISC-sepsis substudy. Patients are included in the ADAPT-sepsis trial based on the following inclusion criteria:

1. Hospitalised adult patients at least 18 years of age;
2. Up to 24 hours since initiation of antibiotics for suspected sepsis;
3. Likely to remain hospitalised and requiring intravenous antibiotics for the next 72 hours;
4. Requirement for critical care.

Patients are excluded based on the following criteria:

1. More than 24 hours since receiving first empiric intravenous antibiotic treatments for suspicion of sepsis;
2. Prolonged (greater than 21 days) antimicrobial therapy mandated (eg, for endocarditis, cerebral/hepatic abscess, tuberculosis, osteomyelitis);
3. Severely immunocompromised (eg, blood neutrophil count less than $0.5\times10^9$/L) not caused by sepsis;

4. Any patient given, or anticipated to receive an IL-6 receptor inhibitor drug (eg, tocilizumab or sarilumab) during their acute hospital admission;
5. All treatment for suspected sepsis likely to be stopped within 24 hours of its initiation because of futility;
6. Patient participation declined;
7. Previously enrolled in ADAPT-Sepsis.

## Outcomes

The primary outcomes are markers of blood leucocyte function measured by flow cytometry. These primary markers are:

► Monocyte HLA-DR
► Neutrophil CD88
► PD-1 on T lymphocytes (CD4-positive and CD8-positive), monocytes and neutrophils
► Percentage of regulatory T cells

Secondary outcome measures include outcome measures derived from the ADAPT-Sepsis trial including duration of antibiotics during the ADAPT-Sepsis intervention phase; total antibiotic days measured at 28 days; occurrence of hospital-acquired infection and length of ICU-stay and hospital-stay. Secondary measures will also include PCT and CRP concentrations in serum.

## Sampling and data collection

As a substudy within the ADAPT-Sepsis trial, all RISC-sepsis procedures and sampling will occur within the trial. Sampling will occur during the ADAPT-Sepsis intervention period when blood is sampled daily. While additional blood samples are taken from patients for RISC-sepsis, there is no increase in number of sampling episodes. Patients will have blood sampled on alternate days up to four sampling points (days 1, 3, 5 and 7). Blood for flow cytometric analysis will be collected in Cyto-Chex (Streck, La Vista, USA) blood preservation tubes before shipping to a central testing hub (Newcastle University). Serum will be collected, frozen and stored at site until the end of the recruitment period.

Baseline clinical data and outcome data will be obtained from the ADAPT-sepsis data collection. Baseline data will include admission diagnosis, site of infection, acute physiology and chronic health evaluation (APACHE) II and sequential organ failure assessment scores and sepsis bundle care elements in the first 24 hours. In addition, at each RISC-sepsis sampling point, the circulating white cell count and differential count will be recorded.

## Blood sample analysis

Flow cytometric analysis will be performed on a single centralised (Newcastle University) BD FACSymphony A5 flow cytometer (Becton Dickinson Biosciences, San Jose, California, USA, from here BDB). Daily internal quality control will be performed using Cytometry Setup and Tracking beads (BDB).

Leucocyte staining will be performed using antibody:-fluorophore conjugates supplied by BDB. Samples will be processed according to a standard operating procedure.

At each time point, serum will be collected for measurement of PCT and CRP. These will be batched and processed by ELISA at the end of the recruitment period.

## Immune phenotyping

Cells will be identified based on light scatter properties and cellular markers: CD15++neutrophils; CD14+monocytes; CD3+CD4+ T cells; CD3+CD8+ T cells; and CD3+CD4+ CD25+ CD127lo regulatory T cells. Expression of monocyte HLA-DR, neutrophil CD88 and PD-1 (CD279) on monocytes, neutrophils and T lymphocytes will be measured.

In addition, exploratory analyses will include: CD14+CD16- classical monocytes; CD14+CD16+ intermediate monocytes; CD14– CD16+non-classical monocytes; CD3– CD19+B cells; CD3– CD56+NK cells; CD3– CD19– CD14– CD16– CD56– HLA–DR+dendritic cells. In addition to expression of HLA-DR and CD279 in cell populations, CD274 (PD-L1) will also be measured.

Data will be assessed for evidence of batch effects and normalisation approaches will be used. Expert manual gating will be performed using a predefined gating standard operating procedure. We will report number and percentage of cell populations in blood and median fluorescent intensity of key markers.

## Definition of sepsis-associated immunosuppression

We will define 'sepsis-associated immunosuppression' as two or more abnormalities (from low HLA-DR, low CD88, raised PD-1 or raised percentage regulatory T cells) that persist for 3 or more days.[14 26] Previous reports have shown that the persistence of immune dysfunctions is associated with worse clinical outcome, including increased hospital-acquired infections and death.[10 13 15] The 'non-immunosuppressed' group will be patients who do not meet these criteria.

## Sample size

The sample size is based on detection of a difference in antibiotic durations between the 'sepsis-associated immunosuppression' and 'non-immunosuppressed' groups. The 2016 Surviving Sepsis Campaign guidelines recommended 7–10 days of antibiotics for patients with sepsis.[27] We have assumed that patients with sepsis-associated immunosuppression will have longer durations (10 days) and so with a SD of 6 days (derived from the ADAPT-sepsis trial), then a sample of 180 patients is required to detect a difference of 3 days in antibiotic treatment between the 'sepsis-associated immunosuppressed' and 'non-immunosuppressed' patients with 90% power, a 5% alpha and a 5% withdrawal rate.

## Data analysis plan

A formal analysis plan will be finalised and deposited on an open access platform before the database is locked.

For the primary objective, the time it takes for antibiotics to be stopped in patients during the ADAPT-sepsis biomarker-guided antibiotic phase will be summarised and the distribution of the outcome described for

the 'sepsis-associated immunosuppression' and the 'non-immunosuppressed' groups. The mean number of antibiotic days will be compared between groups obtained from linear regression models and adjusted for confounding variables. In addition, each of the cellular markers will be assessed as a continuous outcome. The change in cellular markers between each time point will be correlated with the number of days on antibiotics using Spearman's correlation coefficient. One sample t tests will be used to assess the changes between time points.

The trial outcomes of 28-day antibiotics, hospital-acquired infections and length of ICU-stay and hospital-stay will be summarised for the 'sepsis-associated immunosuppressed' and 'non-immunosuppressed' groups. Differences between groups will be assessed using linear regression for continuous variables and logistic regression for categorical variables. We will describe the difference between the groups using p values, point estimates and 95% CIs.

For the secondary objective, we will assess the difference in the PCT/CRP levels and cellular markers for patients who met the criteria for 'sepsis-associated immunosuppressed' and when immune profile recovered (ie, became 'non-immunosuppressed'). Assessments will be captured over time, and we will compute a longitudinal analysis (taking time as a random effect) to assess the difference in the two groups.

Subgroup analyses:

Exploratory analysis will be carried out for the following subgroups:

1. Patients receiving critical care for a medical versus a surgical cause of sepsis;
2. Patients receiving critical care because of trauma versus non-trauma;
3. Patients receiving critical care because of a community-acquired infection versus a hospital-acquired infection;
4. Infection site: community-acquired pneumonia; hospital-acquired pneumonia; urinary tract infection; intra-abdominal infection or positive blood culture;
5. Patients with septic shock, as defined by persistent hypotension requiring vasopressors to maintain an mean arterial pressure (MAP) ≥65 mm Hg and a serum lactate of >2 mmol/L.[1]

Regression models will be used with 'sepsis-associated immunosuppressed' as an interaction term by subgroup for number of days on antibiotics.

## Study duration

The study was planned to commence in April 2020, but due to the COVID-19 pandemic, the start date was delayed until August 2020. Further delays were experienced during the optimisation of flow cytometry and site set up phases. Recruitment commenced in May 2022 and is ongoing. The study is planned to complete in April 2023.

## Public and patient involvement

Members of the public and patients were involved through the Pathfinder group, a group of critical survivors and representatives in the North East of England. The Pathfinder group was involved in developing the research question and lay summary. Ongoing public and patient involvement is through the ADAPT-Sepsis Trial Management and Trial Steering Groups, which incorporate RISC-sepsis oversight, and have members of the Intensive Care Society's Patient and Relatives group.

## Ethics and dissemination

As a substudy, the protocol has been incorporated into the ADAPT-sepsis protocol and ethical approval has been granted by the South Central—Oxford C Research Ethics Committee (IRAS 209815). The study has been registered with the ISRCTN registry (86837685) and adopted by the UK NIHR Clinical Research Portfolio. The study is managed by the Warwick Clinical Trials Unit and Sponsored by The University of Manchester. Independent study oversight is provided by the ADAPT-Sepsis trial steering committee.

The study will be reported in line with the Strengthening the Reporting of Observational Studies in Epidemiology guidelines for observational studies.[28] Findings will be disseminated by peer-reviewed publications and at national and international meetings. In addition, findings will be disseminated via public and patient groups and through social media.

**Author affiliations**
[1]Translational and Clinical Research Institute, Newcastle University, Newcastle upon Tyne, UK
[2]Warwick Clinical Trials Unit, University of Warwick, Coventry, UK
[3]Flow Cytometry Core Facility, Newcastle University, Newcastle upon Tyne, UK
[4]Becton Dickinson, Erembodegem, Belgium
[5]Critical Care Department, Birmingham Heartlands Hospital, Birmingham, UK
[6]Department of Medical Microbiology, Royal Victoria Hospital, Belfast, UK
[7]Wellcome Wolfson Institute for Experimental Medicine, Queen's University Belfast, Belfast, UK
[8]Regional Intensive Care Unit, Belfast Health and Social Care Trust, Belfast, UK
[9]Department of Perioperative Medicine, Freeman Hospital, Newcastle upon Tyne, UK
[10]Intensive Care Unit, Edinburgh Royal Infirmary, Edinburgh, UK
[11]The Queen's Medical Research Institute, The University of Edinburgh, Edinburgh, UK
[12]Integrated Critical Care Unit, South Tyneside and Sunderland NHS Foundation Trust, Sunderland, UK
[13]Division of Immunology, University of Manchester, Salford, Greater Manchester, UK
[14]Critical Care Department, Salford Care Organisation, Greater Manchester, UK
[15]Department of Respiratory Medicine, Royal Victoria Infirmary, Newcastle upon Tyne, UK
[16]JVF Intensive Care Unit, Addenbrooke's Hospital, Cambridge, UK
[17]Division of Anaesthesia, University of Cambridge, Cambridge, UK
[18]Division of Immunology, University of Cambridge, Cambridge, UK
[19]Department of Critical Care Medicine, Royal Victoria Infirmary, Newcastle upon Tyne, UK

**Contributors** The study was conceived by TPH, ACM, MSH, PD, TW, GB and AJS. TPH, ACM, MSH, PD, RMcM, DFMcA, TW, AJS, AR, GP and IMcC acquired research funding. Flow cytometry methods were developed by JS, LT, AF, AR, ACM, MSH, GB, VM and DMcD. The project is managed by HMcN, TE, PM and TPH. Statistical

analysis will be performed by RL, KB and TPH. The draft manuscript was prepared by TPH, ACM, AJS and AF. All authors contributed to reviewing and editing the manuscript.

**Funding** This study/project is funded by the NIHR Efficacy and Mechanism Evaluation (EME) (128374). The views expressed are those of the author(s) and not necessarily those of the NIHR or the Department of Health and Social Care. Regents for flow cytometric analysis are provided by Becton Dickinson Biosciences. The study is also supported by the MRC SHIELD consortium (MRN02995X/1).

**Competing interests** DFMcA is NIHR/MRC EME programme director and has previously sat on NIHR HTA funding committees.

**Patient and public involvement** Patients and/or the public were involved in the design, or conduct, or reporting, or dissemination plans of this research. Refer to the Methods section for further details.

**Patient consent for publication** Not applicable.

**Provenance and peer review** Not commissioned; externally peer reviewed.

**ORCID iDs**
Daniel F McAuley http://orcid.org/0000-0002-3283-1947
Manu Shankar-Hari http://orcid.org/0000-0002-5338-2538
Thomas P Hellyer http://orcid.org/0000-0001-5346-7411

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
