## [Reviewer comments · BMJ Open]

ARTICLE DETAILS

TITLE (PROVISIONAL)	The Role of Immunosuppression in an antibiotic Stewardship intervention and its association with Clinical outcomes and antibiotic use: Protocol for an observational study (RISC-sepsis)
AUTHORS	Scott, Jonathan; Trevi, Loredana; McNeil, Hannah; Ewen, Tom; Mawson, Phil; McDonald, David; Filby, Andrew; Lall, Ranjit; Booth, Katie; Boschman, Gert; Melkebeek, Vesna; Perkins, Gavin; McMullan, Ronan; McAuley, Daniel; McCullagh, Iain J; Walsh, Timothy; Rostron, Anthony; Shankar-Hari, Manu; Dark, Paul; Simpson, A John; Conway-Morris, Andrew; Hellyer, Thomas P

VERSION 1 – REVIEW

REVIEWER	Michael Kueht The University of Texas Medical Branch at Galveston
REVIEW RETURNED	04-Oct-2022

GENERAL COMMENTS	Overall merit-worthy protocol attempting to address an important clinical problem. May benefit from defining "shock" as described in the sub-group analysis. Deferring this change may be acceptable if another document would be a more appropriate location for this information.
--

REVIEWER	Sajal Saha Monash University, Australia, General Practice
REVIEW RETURNED	11-Oct-2022

GENERAL COMMENTS	This study explores the effectiveness of the biomarker-guided antibiotic duration intervention for patients with sepsis-associated immunosuppression. The protocol has been well designed. The study has novelty and implications for antibiotic stewardship in patients with sepsis. There are few points below that need clarifications. Discussion section can be added including limitations of the study. Wondering why neonatal sepsis patients are excluded? Patient exclusion criteria 7 needs clarification in terms of ADAPT sepsis trial, timing. Rationale for alternate days sampling up to 7 days can be explained How and what level public and patients were involved in the study design? "Pathfinder" needs some clarification? Study implications can be discussed in a separate discussion section Study limitations can also be explained under discussion section
---

REVIEWER	Jan Prins Academic Medical Center, Internal Medicine
REVIEW RETURNED	16-Oct-2022

GENERAL COMMENTS	Comments 1. More detail is wanted on the Methods of the main study (ADAPT): what exactly is being done, methods, endpoints. That is important to know as the primary objective is “To determine differences in ADAPT-sepsis trial outcomes between patients defined as ‘sepsis-associated immunosuppressed’ versus ‘non-immunosuppressed” 2. Primary objective is “To determine differences in ADAPT-sepsis trial outcomes between patients defined as ‘sepsis-associated immunosuppressed’ versus ‘non-immunosuppressed”, secondary objective To monitor changes in expression of leukocyte surface markers (primary outcome measures) over the time course of the ADAPT-Sepsis biomarker-guided antibiotic duration intervention period. These objectives are not used consistently throughout the manuscript. For instance, the end of the Introduction (Abstract) says: “which aims to determine the effect of sepsis-associated immunosuppression on antibiotic use”, which is not exactly the same. The study is about biomarkers: the authors already explained that immunosuppression affects antibiotic use. The data analysis plan is also complicated: For the secondary objective, differences in PCT and CRP levels (point estimate and 95% confidence interval) between periods when patients met criteria for ‘sepsis-associated immunosuppressed’ and when immune profile recovered (i.e. became ‘non-immunosuppressed’). This does not align with the secondary objective as stated. This part of analysis plan is in general not very well explained 3. I miss in the Introduction some reflections on the relationship between immunosuppression and biomarkers, which is one of the key points of the study 4. Rationale: Furthermore, if CRP and PCT levels fall over the same period that cellular defects resolve, this would give clinicians greater confidence in the biomarker-guided antibiotic durations. I do not see exactly how that would help me in trusting CRP-guided treatment duration 5. The hypothesis says: “Sepsis-associated immunosuppression will be associated with: longer duration of antibiotics, due to persistently raised PCT and CRP”, which suggests that measuring PCT and CRP would prolong duration of treatment, and not the clinical condition (i.e., the state of immunosuppression) itself 6. How the subset of RISC patients is chosen from the ADAPT trial cohort is not described – we need some information showing that it is a representative sample
--

VERSION 1 – AUTHOR RESPONSE

Reviewer: 1

Dr. Michael Kueht, The University of Texas Medical Branch at Galveston

Comments to the Author:

C1: Overall merit-worthy protocol attempting to address an important clinical problem.

R1: Thank you to Dr Kueht for the positive assessment of the manuscript.

C2: May benefit from defining "shock" as described in the sub-group analysis. Deferring this change may be acceptable if another document would be a more appropriate location for this information.

R2: The definition of 'shock' used is according to the Sepsis-3 definition of persistent hypotension requiring vasopressors to maintain a MAP \geq 65 mmHg and a serum lactate of $>$ 2mmol/L. The manuscript has been revised accordingly.

Reviewer: 2

Mr. Sajal Saha, Monash University, Australia

Comments to the Author:

C3: This study explores the effectiveness of the biomarker-guided antibiotic duration intervention for patients with sepsis-associated immunosuppression. The protocol has been well designed. The study has novelty and implications for antibiotic stewardship in patients with sepsis.

R3: Thank you to Mr Saha for this positive assessment of the protocol.

There are few points below that need clarifications:

C4: Discussion section can be added including limitations of the study.

R4: Thank you for this suggestion. However, the manuscript has been prepared according to the BMJ Open guidelines that does not include a separate 'Discussion' section. Limitations are however covered in the 'Strengths and Weaknesses' section. I will be guided by the Editor as to whether a 'Discussion' section should be added.

C5: Wondering why neonatal sepsis patients are excluded?

R5: The ADAPT-sepsis trial was conceived in response to NICE guidelines (DG18) that concluded that there was insufficient evidence to recommend the routine use of PCT to guide antibiotic durations in the NHS. NICE identified the need to evaluate the use of PCT in critically ill adults and in children. The ADAPT-sepsis trial responded to the unmet need for critically ill adults while another trial, the BATCH trial, is addressing the need in children. For this mechanistic study embedded within ADAPT-sepsis, we are bounded by the trial population, which is adults rather than children.

C6: Patient exclusion criteria 7 needs clarification in terms of ADAPT sepsis trial, timing.

R6: All patients recruited to ADAPT-sepsis are eligible for RISC-sepsis within sites that are participating in the sub-study. There are no additional inclusion and exclusion criteria for RISC-sepsis. The inclusion and exclusion criteria listed relate to ADAPT-sepsis eligibility. Patients can be enrolled to ADAPT-sepsis only once.

C7: Rationale for alternate days sampling up to 7 days can be explained.

R7: Since the purpose of the study is to *explain* the role of sepsis-associated immunosuppression on trial outcomes and the changes in immune phenotype over time with PCT/CRP levels, serial sampling is required. Alternate day sampling was a pragmatic decision to allow sampling that spanned the ADAPT-sepsis intervention period (on average a week), while reducing the costs of daily sampling.

C8: How and what level public and patients were involved in the study design? "Pathfinder" needs some clarification?

R8: Thank you for raising this. I have provided more information in the manuscript. We worked with the Pathfinder group to develop the research plan prior to funding application, which was before the

COVID pandemic. During the COVID pandemic, the Pathfinder group disbanded and no longer provided PPI input. However, RISC-sepsis is incorporated into the trial management and trial steering groups for ADAPT-sepsis and so PPI oversight continues through those groups, as it does for the main trial.

C9: Study implications can be discussed in a separate discussion section Study limitations can also be explained under discussion section

R9: As per my earlier response, I will defer to the BMJ Open editorial team regarding the addition of a 'Discussion' section. However, I believe that the implications of the study are addressed in 'Rationale' section.

Reviewer: 3

Dr. Jan Prins, Academic Medical Center

Many thanks to Dr Prins for this review.

Comments to the Author:

C10: More detail is wanted on the Methods of the main study (ADAPT): what exactly is being done, methods, endpoints. That is important to know as the primary objective is "To determine differences in ADAPT-sepsis trial outcomes between patients defined as 'sepsis-associated immunosuppressed' versus 'non-immunosuppressed'"

R10: Additional information is added paragraph 3 of the 'Introduction' to provide information on trial outcomes for ADAPT-sepsis. While it is important to provide the Reader with sufficient information about the ADAPT-sepsis trial to understand the RISC-sepsis study (as requested by Dr Prins), I want to avoid confusing the Reader by providing too many details of ADAPT-sepsis. I hope I have struck the right balance.

C11: Objectives are "To determine differences in ADAPT-sepsis trial outcomes between patients defined as 'sepsis-associated immunosuppressed' versus 'non-immunosuppressed'" and "To monitor changes in expression of leukocyte surface markers (primary outcome measures) over the time course of the ADAPT-Sepsis biomarker-guided antibiotic duration intervention period". These objectives are not used consistently throughout the manuscript. For instance, the end of the Introduction (Abstract) says: "which aims to determine the effect of sepsis-associated immunosuppression on antibiotic use", which is not exactly the same. The study is about biomarkers: the authors already explained that immunosuppression affects antibiotic use.

R11: I have revised the 'Introduction' section of the abstract to remove this inconsistency, stating "This mechanistic study aims to determine the effect of sepsis-associated immunosuppression on the trial outcome measures". This is consistent with the primary objective, allowing the 'Primary objective' section to elaborate further.

C12: The data analysis plan is also complicated: For the secondary objective, differences in PCT and CRP levels (point estimate and 95% confidence interval) between periods when patients met criteria for 'sepsis-associated immunosuppressed' and when immune profile recovered (i.e. became 'non-immunosuppressed'). This does not align with the secondary objective as stated. This part of analysis plan is in general not very well explained.

R12: Thank you for raising this. The section of the 'data analysis plan' has now been revised and more clearly aligned with the secondary outcome.

C13: I miss in the Introduction some reflections on the relationship between immunosuppression and biomarkers, which is one of the key points of the study.

C14: Rationale: Furthermore, if CRP and PCT levels fall over the same period that cellular defects resolve, this would give clinicians greater confidence in the biomarker-guided antibiotic durations. I do not see exactly how that would help me in trusting CRP-guided treatment duration.

C15: The hypothesis says: "Sepsis-associated immunosuppression will be associated with: longer duration of antibiotics, due to persistently raised PCT and CRP", which suggests that measuring PCT and CRP would prolong duration of treatment, and not the clinical condition (i.e., the state of immunosuppression) itself.

R 13-15: In response to C13-15, the 'Introduction' has been revised to give more detail on the cellular markers that identify the immune phenotype of 'sepsis-associated immunosuppression'. Importantly references have been added that link PCT and CRP to sepsis-associated immunosuppression (predominantly mHLA-DR), which provides greater context for the Rationale and Hypothesis.

C16: How the subset of RISC patients is chosen from the ADAPT trial cohort is not described – we need some information showing that it is a representative sample.

R16: Sites participating in ADAPT-sepsis are able to join ADAPT-sepsis depending on their site capacity to deliver both studies. Sites that are participating in both RISC-sepsis and ADAPT-sepsis include sequential patients such that the RISC-sepsis population is representative of the overall trial population (see 'Methods and analysis' / 'Population').